# Spatial Omics Imaging of Fresh-Frozen Tissue and Routine FFPE Histopathology of a Single Cancer Needle Core Biopsy: A Freezing Device and Multimodal Workflow

**DOI:** 10.3390/cancers15102676

**Published:** 2023-05-10

**Authors:** Miriam F. Rittel, Stefan Schmidt, Cleo-Aron Weis, Emrullah Birgin, Björn van Marwick, Matthias Rädle, Steffen J. Diehl, Nuh N. Rahbari, Alexander Marx, Carsten Hopf

**Affiliations:** 1Center for Mass Spectrometry and Optical Spectroscopy (CeMOS), Mannheim University of Applied Sciences, Paul-Wittsack-Str. 10, 68163 Mannheim, Germany; 2Institute of Medical Technology, Heidelberg University and Mannheim University of Applied Sciences, Paul-Wittsack-Str. 10, 68163 Mannheim, Germany; 3Medical Faculty Mannheim, Heidelberg University, Theodor-Kutzer-Ufer 1-3, 68167 Mannheim, Germany; 4Institute of Pathology, University Medical Centre Mannheim, Theodor-Kutzer-Ufer 1-3, 68167 Mannheim, Germany; 5Department of Surgery, University Medical Centre Mannheim, Theodor-Kutzer-Ufer 1-3, 68167 Mannheim, Germany; 6Clinic of Clinical Radiology and Nuclear Medicine, University Medical Center Mannheim, Theodor-Kutzer-Ufer 1-3, 68167 Mannheim, Germany

**Keywords:** MALDI imaging, biopsy, multimodal, spatialomics, infrared, immunohistochemistry, co-registration, mass spectrometry

## Abstract

**Simple Summary:**

Routine clinical approaches for cancer diagnosis demand fast, cost-efficient, and reliable methods, and the implementation of these methods within clinical settings. Currently, histopathology is the gold standard for tissue-based clinical diagnosis. Recently, spatially resolved molecular profiling techniques such as mass spectrometry imaging (MSI) or infrared spectroscopy imaging (IRI) have increasingly contributed to clinical research, e.g., by differentiating cancer subtypes using molecular fingerprints. However, the adoption of the corresponding workflows in clinical routines remains challenging, especially for fresh-frozen tissue specimens. Here, we present a novel device based on 3D-printing technology, which facilitates the sample preparation of needle biopsies for correlated clinical tissue analysis. It enables the use of a combination of MSI and IRI on fresh-frozen clinical samples, with a histopathological examination of the same needle core after formalin fixation and paraffin embedding (FFPE). This device and workflow can pave the way for a more profound understanding of biomolecular processes in cancer and, thus, facilitate more accurate diagnosis.

**Abstract:**

The complex molecular alterations that underlie cancer pathophysiology are studied in depth with omics methods using bulk tissue extracts. For spatially resolved tissue diagnostics using needle biopsy cores, however, histopathological analysis using stained FFPE tissue and the immunohistochemistry (IHC) of a few marker proteins is currently the main clinical focus. Today, spatial omics imaging using MSI or IRI is an emerging diagnostic technology for the identification and classification of various cancer types. However, to conserve tissue-specific metabolomic states, fast, reliable, and precise methods for the preparation of fresh-frozen (FF) tissue sections are crucial. Such methods are often incompatible with clinical practice, since spatial metabolomics and the routine histopathology of needle biopsies currently require two biopsies for FF and FFPE sampling, respectively. Therefore, we developed a device and corresponding laboratory and computational workflows for the multimodal spatial omics analysis of fresh-frozen, longitudinally sectioned needle biopsies to accompany standard FFPE histopathology of the same biopsy core. As a proof-of-concept, we analyzed surgical human liver cancer specimens using IRI and MSI with precise co-registration and, following FFPE processing, by sequential clinical pathology analysis of the same biopsy core. This workflow allowed for a spatial comparison between different spectral profiles and alterations in tissue histology, as well as a direct comparison for histological diagnosis without the need for an extra biopsy.

## 1. Introduction

Cancer diagnosis often requires a biopsy, i.e., the extraction of small, localized tissue samples from patients using elaborate mechanical tools. Personalized therapy using interventional radiology and/or adjuvant pharmacotherapy increasingly requires advanced molecular tissue analytics, e.g., for tumor sub-classification, which can aid decision making by tumor boards. Complementing non-invasive clinical diagnostic imaging, biopsies allow for the direct investigation of the heterogeneous molecular composition and complex pathophysiology of a tissue with high cellular specificity. A core needle biopsy is a procedure that is frequently used for the direct examination of neoplastic lesions such as soft-tissue sarcoma [1], breast cancer [2] or liver neoplasms [3], and for ancillary molecular testing. Nowadays, pathologies of the liver, such as fibrosis, cirrhosis, or cancer, are routinely addressed using a percutaneous needle core biopsy [4]. Although this method is widely adopted in oncology and beyond, the existing devices and procedures are not yet designed to support modern spatial omics technologies [5]. In routine pre-omics-era clinical practice and in accordance with clinical guidelines [6], the examination of resected tissues or biopsies is still largely performed based on expert pathological interpretation of the histomorphological features of the stained tissues, albeit in an increasingly computational fashion [7]. In addition, the analysis of cell heterogeneity and spatial organization within the tissue can be aided by the immunohistochemistry (IHC) of a rather small number of marker proteins. In contrast, mass-spectrometry- (MS-)based omics methodologies have recently benefitted from vastly improved sensitivity [8], a trend that has prompted the development of miniaturized workflows with primary applications in the analysis of bulk extracts from core needle biopsies [9,10].

With the advent of spatially resolved omics technologies, such as the MSI of proteins, lipids, and metabolites (“spatial lipidomics/metabolomics”) [11,12,13,14,15], MSI-guided micro-proteomics [16], microscopy-guided micro-proteomics [17], spatial transcriptomics, and multi-omics [18,19] for studies of complex molecular alterations in pathophysiology in tissue specimens, their potential to be used in clinical diagnosis, prognosis, or targeted therapy is manifold. They allow for the investigation of intra-tumor heterogeneity and pathophysiological processes including tumor genesis, progression or metastasis. In all of these processes, spatial information is essential. Within the field of spatial omics, MSI provides high speeds and high chemical specificity at a cellular spatial resolution [11,12,13]. Matrix-assisted laser desorption/ionization (MALDI) MSI, in particular, has triggered clinically relevant research on multiple aspects of liver disease, ranging from colorectal cancer metastasis [20] to drug distribution in liver metastases [21] to hepatocellular carcinoma [22]. It is also considered a promising technology for next-generation clinical pathology practice [23].

Studies often combine complementary imaging techniques to determine the complex biomolecular context of a sample in sophisticated so-called multimodal workflows [24,25,26,27]: label-free imaging techniques that can be used on the same tissue section prior to MSI are infrared imaging (IRI) and IR microscopy. Due to recent technological innovations in vibrational spectroscopy, IR methods enable the rapid screening and examination of tissue sections for tumor margin detection, classification, and grading [25,28,29,30,31]. Similarly, to correlative histology-based MSI approaches [32], the morphological features obtained with IR spectroscopy yield accurate labels for precise region-specific MSI-based molecular investigations [25], a procedure known as “IRI-guided MSI”. However, correlating different spatial omics approaches requires data analysis pipelines and strategies in the fields of sample preparation, image fusion, and image co-registration and data evaluation. Similar requirements are mandatory for direct comparisons with FFPE-based histopathology, which is the current gold standard.

Whereas clinical routine analysis and spatial proteomics can be performed on formalin-fixed paraffin-embedded (FFPE) tissue sections, lipidomic and metabolomic approaches, including IRI and MSI, favor fresh-frozen (FF) material to reveal unmodified molecular signatures. Although several studies have shown that some metabolites are preserved in FFPE tissue for MSI analysis, limitations in preserving the unmodified molecular content are generally recognized [33,34]. In addition, to conserve tissue-specific metabolomic states, fast, reliable, and precise methods for the preparation of fresh-frozen tissue sections are crucial. They are often challenging to implement in clinical practice, however, especially regarding the need for longitudinally cut biopsy sections to yield a sufficient cross section for analysis. The processing of FF biopsy samples is difficult in terms of collection, storage, and cryo-sectioning. In addition, an optimized freezing protocol is mandatory to minimize morphological alterations that occur due to freezing damage. Therefore, a standardized procedure that limits environmental and human errors is advantageous. The available sample preparation protocols and tools [35] were developed for FFPE-based tissue embedding and are inconsistent with requirements for lipid/metabolite MSI. In addition, common methodologies, such as freezing biopsies directly onto a metal chuck, involve sample transport in saline, during which the molecular state of the sample may be altered. Therefore, core biopsies are difficult to handle with this technique, especially for non-continuous cores or for ambitious multimodal approaches. In a classic study, Cazares et al. (2009) presented a combined analysis of longitudinally cut, mirrored biopsy cores by MSI and histology [36]. Shiraishi et al. (2020) recently presented an advanced device to divide a biopsy into two similar parts [37]. However, it lacks the ability to snap-freeze the sample and to analyze both FF and subsequent FFPE tissues from the same needle core.

To overcome the challenges and limitations for the use of core needle biopsies in the field of spatial omics, a device for the biopsy FF sample preparation of longitudinally sectioned biopsies was developed. Clinical requirements such as sterilizability, cost, and ease-of-handling were also considered in the design. It allows for a reliable and robust multimodal workflow to combine the IRI and MALDI imaging of FF tissue with routine histopathological FFPE tissue analysis of a single biopsy core. Further, it enables handling of minimal invasive, non-continuous biopsy cores for any analyses requiring FF sample preparation. Multiple sections could be retrieved for different analyses, and accurate image co-registration was employed for direct comparison. Using this workflow, the correlation between differentially expressed spectral profiles and the tissue histology of FF material was investigated. Furthermore, a comparison between pathological examination of FFPE and molecular or spectral profiles was performed. This study presents the design of the device and the 3D printing manufacturing process. The corresponding workflow is described in detail. Furthermore, a feasibility study with two patients (diagnoses: hepatocellular carcinoma (HCC) and mamma metastasis in the liver) and two spectral measurement techniques (MSI and Fourier transform (FT)-IRI) on FF tissue, as well as the clinical routine histopathological assessment of a subsequent FFPE preparation, all conducted on the same biopsy core, are presented.

## 2. Materials and Methods

### 2.1. Tissue Specimens and Needle Biopsies

Human tissue specimens were collected from excised tumors during surgery at the Mannheim University Clinic. Details of the ethics assessment can be found in the ethics statement. Following the published guidelines on the use of liver biopsies in clinical practice [6], biopsies were excised using a core biopsy instrument (gauge size of 18G, penetration depth of 22 mm; Bard Peripheral Vascular Inc., Tempe, AZ, USA). The specimens analyzed in this study included one HCC (female, 73 years of age) and one mamma metastasis residing in the liver (female, 31 years of age). Three biopsies were taken for the HCC sample and two were taken for the mamma metastasis sample due to the size limitations of the nodule. One biopsy per tumor entity is shown as a representative example in the main manuscript; all others are depicted in the Appendix A. The initial experiments were conducted on biopsies punched from a bovine liver obtained from a local butcher.

### 2.2. Biopsy Freezing Device, Preparation of Fresh-Frozen Samples

An Autodesk Inventor Professional 2022 (Autodesk Inc., Mill Valley, CA, USA) was used to compile the CAD designs of all the required parts. The base compartment, which was made from metal, was directly printed as a positive module using austenitic steel powder (MetcoAdd 316L-A, Oerlikon Metco Europe GmbH, Raunheim, Switzerland) and a 3D metal printer (TruPrint 2000, Trumpf, Stuttgart, Germany). Silicone parts comprising a base and a top compartment were retrieved from negative casting molds (Figure 1b).

Negative casting molds were fabricated from a biocompatible photopolymer resin (FLDGOR01, Formlabs, Berlin, Germany) by low force stereolithography using a 3D-printer (Form2, Formlabs, Berlin, Germany). Prior to use, the negative casting molds were washed with ethanol in an ultrasonic cleaner and subsequently dried at room temperature and under ambient pressure. The two-component silicone rubber (REPLISIL 32 N, Silconic, Lonsee, Germany) used for the base and cover was used in accordance with manufacturer’s recommendations; it consisted of a two-component system (manufacturer components, solution A: solution B, 1:1 (*v*/*v*)) and was poured into the casting molds directly after mixing. The forms were vulcanized at room temperature (RT) and the silicone curation took about 20 min. In order to use the tool for FF sample preparation, punched needle biopsies were skimmed off onto the elevation on the base compartment, henceforth referred to as the bridge, which was then covered with the top compartment and filled with the embedding medium hydroxypropyl methylcellulose: polyvinylpyrrolidone (HPMC:PVP, 1:1, *w*/*w*) [38]. In brief, the embedding medium was prepared by dissolving 2.5% PVP and 7.5% of HPMC in de-ionized water dH_2_O (prepared in house); it was then stirred in an ice bath overnight and kept at 4 °C for several hours. For long-term storage, the hydrogel was kept at −20 °C in a freezer. For snap-freezing, the biopsy-loaded device was then placed on an aluminum foil floating on liquid nitrogen. When frozen, the samples were stored at −80 °C until use. For sectioning, the samples were equilibrated in a cryostat (CM1860 UV, Leica Biosystems, Nussloch, Germany) at −16 °C for about 20 min. The flexible top compartment of the device was then removed and the sample was detached with pre-cooled forceps and mounted upside-down onto a metal chuck for cryo-sectioning (10 µm section thickness); it was later used for spatial omics analyses and H&E reference staining of the FF sections. A video showcasing the entire procedure is provided with the Appendix A.

### 2.3. MALDI Mass Spectrometry Imaging (MSI) and FT-IR Imaging (IRI)

For MSI and IRI, tissue cryosections were thaw-mounted onto indium tin-oxide-coated microscope slides (ITO slides; 8–12 Ohm resistance, Diamond Coatings Ltd., Brierley Hill, UK). Following 15 min equilibration at RT, bright field (BF) optical reference images were acquired using an Aperio CS2 scanner (objective: 20×/0.75 NA Plan Apo, Leica Biosystems, Nussloch, Germany) for the elucidation of tissue positions in MSI. Subsequently, IRI was performed on a Spotlight 400 FT-IR Imaging System (Perkin Elmer LAS, Rodgau, Germany) as described elsewhere [25]. In brief, images were recorded with a spatial resolution of 25 µm with a wave number range of 750–4000 cm^−1^ at 4 cm^−1^ spectral resolution, with 2 scans per pixel and 2.2 cm/s mirror speed. Spectra were preprocessed to common standards (2nd derivative and standard normal variate (SNV) normalization) [39], and images were visualized using specio, a freely available python package [http://imageio.github.io (last accessed on 5 May 2023)]. MSI was performed on the same section after IRI. For MSI, slides were coated with a 1,5-diamino-naphthalene (DAN) (D21200, Sigma-Aldrich, Taufkirchen, Germany) matrix, as described elsewhere [40]. In short, 1,5-DAN at 10 mg/mL in a 50% acetonitrile/H_2_O (*v*/*v*) was sprayed onto the sample using an HTX TM-Sprayer (HTX Technologies, Chapel Hill, NC, Vereinigte Staaten). The matrix was deposited in 10 layers at a flow rate of 100 µL/min and a spray-head velocity of 1200 mm/min with a distance of 3 mm between the sprayed lines (HH pattern) and a spray nozzle height of 40 mm; the temperature was 60 °C, the pressure was 10 psi, and the gas flow rate was 2 L/min. Measurements were performed on a 7 T XR magnetic resonance (MR) mass spectrometer (solariX, Bruker Daltonics, Bremen, Germany) in negative ion mode in the range of *m*/*z* 200–2000, at a lateral step-size of 50 µm, using a 1 M transient (free induction decay time of 0.98 s). On-line calibration was performed using the ubiquitous endogenous [M-H]^−^ ion of PI 38:4 (*m*/*z* 885.5487). For each mass spectrum, ions from 200 shots at 2000 Hz were accumulated at a laser power of 25%. Prior to analysis, the system was calibrated using the ESI source with the ESI-L low-concentration tuning mix (G1969-85000, Agilent Technologies, Waldbronn, Germany). Mass spectra were visualized using SCiLS Lab 2023a Pro (Bruker Daltonics, Bremen, Germany). The mass-search window of the *m*/*z* values was inferred from the mass resolution specific to that measurement and that device based on the full width at the half-maximum (FWHM; resolving power of ~135,000 at *m*/*z* 400, ~73,500 at *m*/*z* 800) of the corresponding mass peak, which was determined as described. The MSI results were evaluated using discriminant ROC analysis and visualized non-normalized using SCiLS Lab Version 2023a Pro (Bruker Daltonics, Bremen, Germany). Molecular annotation was performed using Metaspace (https://metaspace2020.eu/ (last accessed on 5 May 2023)).

### 2.4. Precise Co-Registration

Spectral images from MSI and IRI were co-registered with reference pathology H&E analysis. To develop the registration workflow and assess the registration quality of the longitudinally sectioned biopsy samples, bovine liver samples were used. For registration, the python packages SimpleITK [41] and simpleElastix [42] were used. With the latter, image registration was performed by sequential rigid, affine, and b-spline transformations. Advanced Mattes mutual information was used as a metric for the linear interpolator [43]. In total, about 1000 iterations were performed for each linear transformation and about twice as many iterations were performed for the non-linear transformations.

The registration methods were tested and evaluated on a bovine liver data set consisting of three needle biopsy cores, and they were then applied to the human dataset. The testing, optimization, and validation of the image registration parameters were performed on adjacent H&E-stained bovine liver tissue sections. For visual comparison prior to image registration, color normalization of the two H&E-stained histological images was performed by aligning the color distributions of each color channel between the two images. To this end, morphological features including blood vessels were selected as landmarks based on the manual examination of the tissue, and they were annotated using the Aperio Image Scope software (Leica Biosystems, Wetzlar, Germany). Two different parameters were obtained to assess registration quality: a point-to-point comparison of corresponding landmarks, the distances of the center of gravity of the annotated regions and Sorensen–Dice similarity [44]. Briefly, the formula used to calculate the Dice similarity is: Dice = 2 × overlapping area/sum of both corresponding areas. For each biopsy, six regions of interest (ROIs) that were evenly distributed over the biopsy were selected. Both parameters were calculated as a function of the relative distance between individual tissue sections. The results are presented as box plots, with the box marking the first and third quartiles, as well as the median. Whiskers represent the highest value within the 1.5× inter-quartile range (IQR). The results were calculated both for all landmarks of the 3 biopsies in a total of 3 biopsies × 6 landmarks in 1 plot, and also for each biopsy individually, meaning 6 landmarks per biopsy. In both cases, a second-order polynomial function was fitted to the data to guide the eye. The registration results were calculated and plotted using Python. Graphics were assembled using power point (Microsoft Corporation, Redmond, DC, USA) and Inkscape (GNU General Public License, version 3).

### 2.5. Conversion of the Remaining FF Samples to FFPE and Clinical Pathology Analysis

After cryo-sectioning, the remainder of each FF biopsy in the embedding medium was immersed in 5 mL of 10% phosphate-buffered formalin (Sarstedt, Nürmbrecht, Germany) at RT. The non-dissolved embedding medium can be removed during this step when floating at the top of the solution if desired. After at least 22 h, the biopsies were embedded in paraffin according to the clinical routine FFPE workflow. FFPE tissue sections (2 µm thickness) obtained with a Leica RM2245 rotary microtome (Leica Biosystems, Wetzlar, Germany) were mounted on glass slides (Superfrost Plus, Epredia, Braunschweig, Germany). Staining details for the FF- and FFPE-prepared sections are described in the Appendix A.

## 3. Results and Discussion

Solid tumors typically require biopsies for accurate diagnosis and for the selection of the optimal treatment for individual patients. So far, only limited spatial omics research has been conducted on biopsy cores, although proteomics, metabolomics, and MSI have become mainstays in cancer research [8,12]. This is due to obstacles including insufficient handling capabilities for FF biopsy core materials [4,45], which is optimal for many omics techniques. Nevertheless, clinical research consortia, such as the Mannheim Molecular Intervention Environment (M^2^OLIE; http://www.m2olie.de/en/ (last accessed on 5 May 2023)), which focus on innovative tumor therapies using molecular intervention, rely on the diagnosis of core biopsy material obtained in interventional radiology for diagnosis. For this reason, a device and corresponding laboratory and computational workflows were developed for the handling of FF biopsy cores and for subsequent incorporation into clinical FFPE routines. Multiple sections can be retrieved for multimodal spatial omics analysis, and the remaining cores can be further processed into FFPE specimens for subsequent clinical routine histopathology analysis.

### 3.1. A 3D-Printed Flexible Embedding and Freezing Device for the Longitudinal Sectioning of Needle Biopsies

In order to facilitate the parallel clinical use of single core needle biopsies in spatial multi-omics studies that require FF tissue, as well as in routine histopathology that utilizes FFPE tissue, we first set out to design and manufacture a new device for the fast and robust embedding, freezing, and longitudinal sectioning of core biopsies (Figure 1). The device was designed to ensure functionality and clinical compatibility (Figure 1a); this includes the overall sizing and notch at the back (1), for which the entire device fits into commercially available embedding cassettes and allows them to be closed for routine labelling, e.g., barcoding and storage. Furthermore, the u-shaped mold (2) formed by the cover enables the precise orientation of a parallel sectioning plane between the longitudinal plane of the biopsy core and the blade before cryo-sectioning. This reduces sample loss due to faster accurate adjustment (Appendix A). Notably, a tilted sample would result in sectioning that was not exactly parallel to the longitudinal plane of the biopsy, thus resulting in a smaller measurement area. In addition, the diameter of a biopsy core is very small, usually well below 1 mm, and tissue sections can only be taken from a reasonable depth of the sample (Appendix A).

Two different base materials (3), metal and silicone, which have different thermal conductive properties, were tested for sample freezing damage and showed similar results (Appendix A). Both can be sterilized and, in principle, re-used. Furthermore, both materials are inert and do not react in any way with the sample. Other solutions for aiding biopsy sample preparation include filtering paper [37], which might induce the diffusion of liquids out of the sample and therefore change the spatial distribution of small molecules. The bridge (4) in the proposed design facilitates the transfer of the biopsy core from the needle to the device. This is especially important for biopsy needles that include a notch, in which the biopsy resides after excision. Without a punch within the cannula, the biopsy cannot be pushed out of that cannula onto any assisting device. Due to the indentation in which the samples resides, it is impossible to transfer the biopsy onto a flat surface. For this reason, the bridge fits into the indentation of the needle and facilitates the release of the biopsy sample onto the device. The cover is made from silicone, which remains bendable and thus removable at negative temperatures (5). Hence, it allows for convenient handling while the samples remain frozen in the cryostat. The use of solid materials such as plastics or metal resulted in the sample becoming jammed in the cover, and it could not be removed without inducing damage to the sample. The lessons learned during device development are summarized in Appendix A. The final design consisted of two negative casting molds made from resin for the positive silicone parts (Figure 1b).

### 3.2. Sample Preparation Workflow for Multimodal Multi-Omics Analysis of Needle Biopsies

To effectively use the embedding and freezing device, we next developed a sample preparation workflow for consecutive FF and FFPE preparation. The process included the following steps (Figure 2a): after excision of the biopsy, the core was skimmed off onto the bridge of the device’s baseplate (i). By placing the silicon cover on top of the base, a mold was formed around the bridge that can be filled with an MSI-compatible embedding medium (ii). Embedding also enables the handling of non-continuous cores, whereby no part of the sample is lost. The sample can then be snap-frozen by floating the device in aluminum foil on liquid nitrogen (iii). Before FF biopsy cryo-sectioning, the device was equilibrated in the cryostat, after which the cover was removed (iv) while the sample remained frozen. Afterwards, the sample was removed from the base and mounted upside-down on a metal chuck for cryo-sectioning (v). Collected sections can then be used for IRI and MSI (vi), which require FF sample preparation. In addition, one FF tissue section was H&E-stained for reference analysis. A tutorial video on the handling of the device for FF preparation up to spectral analyses is provided in the Appendix A. Following cryo-sectioning, the remaining sample was detached from the metal chuck and dissolved in formalin for fixation at RT and for transfer to the pathology without sample damage (vii). Thereafter, routine clinical histopathology analysis including paraffin embedding (viii), sectioning (ix), and subsequent staining was carried out (x) (Figure 2a).

For the multimodal spatial omics analysis of tissue, several adjacent tissue sections are required. Using the device and workflow described above, we retrieved at least 8 adjacent sections of 10 µm thickness for analyses performed on FF tissue and, after subsequent FFPE preparation, at least another 8 sections of 2 µm thickness for analyses of the FFPE-prepared sections from core biopsies with a diameter of about 800 µm.

For the proof-of-concept work, biopsy needle core samples were initially punched from a bovine liver. Bright field (BF) reference images, MSI data, and H&E-stained images were obtained from FF sections of bovine liver biopsies. H&E, Berlin blue, and May–Grünwald staining were performed on the FFPE sections (Figure 2b). This procedure was conducted for two biopsy cores. Multiple additional adjacent FF sections were H&E-stained to visualize the tissue’s morphological quality and the similarity of adjacent sections, and to assess the registration quality between different modalities for later data fusion (Appendix A). Subsequently, these FF-first needle cores were prepared with FFPE. The tissue quality and morphology were indistinguishable from a needle core that was directly FFPE-processed, i.e., without prior FF procedures. It was therefore deemed suitable for accurate pathological annotations (Appendix A). Using this approach, six out of six biopsies obtained from human samples could be successfully prepared for analysis on FF as well as FFPE sections.

This demonstrated the robustness of the workflow. Moreover, it enabled the use of multimodal spatial omics technologies on longitudinally sectioned needle biopsies alongside clinical routine analysis. This approach has the advantage that FF and FFPE analyses are comparable and do not have to be performed on two different biopsy cores, which are difficult to retrieve from the same patient for clinical and ethical reasons, and which might not contain the same tissue due to intra-tumor and general tissue heterogeneity.

### 3.3. Image Co-Registration for Multimodal Spectral Analyses of Needle Biopsies

Multimodal analysis often requires different sections for each type of analysis, e.g., MSI, spatial transcriptomics, etc., since the techniques are often destructive [12,46]. One common morphological reference for regions of interest (ROIs) is provided by H&E-stained tissue sections with expert pathological annotations. Since sectioning is still mostly a manual process, errors such as the deformation of adjacent sections frequently occur, especially in small samples such as biopsy cores. To be able to combine results from different sections, one image, the moving image, needs to be computationally transformed to fit the other fixed image, in a process called image co-registration. Usually, microscopic bright field (BF) images used for positional referencing in omics techniques are registered with an adjacent H&E-stained ROI reference section. A checkerboard representation of the registration results of two adjacent sections, one H&E and the second BF, provides a quality estimate of the registration accuracy (Figure 3a). Since non-stained microscopic images do not reveal enough structural details for the assessment of registration quality, multiple adjacent bovine liver sections were H&E stained. Arteries were annotated as landmark regions of interest (ROIs) in each individual section. They are the most recognizable morphological features in liver tissue and can usually be followed across several adjacent sections. These ROIs could also be used to determine the overlap between the two registered images and morphological structures as a quality measure for the registration result.

A checkerboard representation visualized the results of the two different registration methods, (i) linear, e.g., rigid, and (ii) non-linear, e.g., b-spline, for two H&E-stained sections (Figure 3b). Overlays of the biopsy cores after rigid transformation revealed local tissue deformations arising from the sectioning process. Non-overlapping ROIs and deviations of the outer shapes of the biopsy cores were observed, as simple rigid registration does not account for local deformations (Figure 3b(i)). Consequently, they could not be matched with this type of transformation method. Accordingly, the ROIs of the fixed section image (yellow) and the moving section image (blue) did not overlap (green) well. In contrast, when using non-linear registration (b spline), which allows for local corrections including initial linear pre-alignment registration steps, the landmarks and overall shape of the biopsy core were spatially aligned (Figure 3b(ii) and Appendix A).

Accuracy of image co-registration depends on the chosen metric. Two different metrics for registration quality were calculated for each registration. One was a positional parameter, the median Euclidean distance of the center-of-gravity labelled the “center distance” of the annotated features [44,47]. It describes the distance between the ROIs after registration. The Euclidian distance, however, neglects the uniform enlargement of the ROIs between two sections, as long as the center of gravity remains the same, which describes more precisely the fit of the position of the two ROIs after registration. The second metric, the Sorensen–Dice coefficient [44] (SDC), which evaluates areas rather than positions and reflects the similarity of two given shapes according to their relative spatial distribution, also takes deformations of the ROIs into account.

For multimodal approaches, more than two sections, sometimes adjacent, sometimes with multiple sections in between, need to be registered for comparisons between multiple modalities. For example, tissue sections #3 to #8 were registered as moving images relative to section #1, the fixed image (Figure 3c). This reflected different numbers of sections that were left out between co-registered sections, e.g., a distance of 2 for #1–#3, a distance of 4 for #1–#5, and so on. Sections #2 and #4 were not H&E stained but were used for MSI instead. An artery that exhibited morphological changes throughout the biopsy core was chosen as an exemplary landmark (Figure 3c). The corresponding center distances and SDCs between the landmarks of the moving (#3–#8) and fixed images (#1) were calculated. Different types of sectioning artefacts, such as discontinuous cores, longitudinal stretching of the entire biopsy, or bending of the core, as well as other examples of annotated arteries and morphological features, which underwent less pronounced change throughout the eight adjacent sections, are given in the Appendix A. These individually calculated statistics for registration accuracy allow for a detailed assessment of each individual section and of outliers of the manual sectioning process. In contrast, a summary statistic as depicted in Figure 3d enables the overall evaluation of co-registration precision across several replicates.

Morphological changes of shape across several adjacent sections and thus the center position of the ROIs and deformations of the entire tissue section, as well as artefacts introduced during cryo-sectioning, significantly impact registration accuracy. For example, some morphological features such as arteries barely change over the eight adjacent sections, whereas others undergo significant positional change (Appendix A). Such effects are not accounted for in linear image co-registration (Appendix A) and are considered only to a certain extent in non-linear transformation. Since the registration used here is based on intensity, the algorithm markedly transforms only those morphological features whose intensity distribution differs from their surroundings, as the tissue does from the background. To give one example, the best scores for both metrics were achieved for a landmark that presumably represented the cross-section of a vessel (Appendix A). The high scores of this landmark can likely be attributed to the fact that its shape was nearly constant throughout the eight adjacent sections and that the contrast between the background and the surrounding tissue was high. With the increasing relative distance of the sections, the center distances increased and the SDCs decreased, confirming that, unsurprisingly, co-registration was more precise for neighboring tissue sections (Figure 3d). For small distances between adjacent tissue sections (1–3 sections rel. dist.), the median center distance was below 50 µm, i.e., below the size of one MSI pixel. A detailed overview of the results for different biopsy samples is presented in Appendix A.

### 3.4. Clinical Proof of Concept of FF and FFPE Tissue Multimodal Spatial Omics Analysis of the Same Needle Biopsy Core

To demonstrate the applicability of the proposed device and workflow in a clinical context, we obtained samples from two patients: one diagnosed with grade 2 hepatocellular carcinoma (HCC) and one with an invasive lobular mamma carcinoma metastasis (MammaMet) residing in the liver. Biopsies were not obtained from the patients directly, but from freshly resected tissue samples during surgery. The extracted core biopsies (3× HCC, 2× MammaMet) were embedded and frozen using the device and processed via the proposed workflow (Figure 2a). For clinical proof of technical feasibility, we aimed to combine the modalities of MSI, IRI, and H&E staining conducted on FF tissue sections and to subsequently reference them against histopathological H&E and IHC staining performed on FFPE sections from the same needle biopsy core (Appendix A). Multimodal correlative imaging using MSI, IRI, and other modalities has been reported by several laboratories [24,25,27,28,48]. Despite the obvious advantages of using a single-biopsy-analysis for patients and for analytical results, to our knowledge, it is not yet possible to undertake multimodal spatial omics imaging and classical histopathology together from the same core needle biopsy that is processed successively: FF first, then FFPE.

As in the initial experiments with bovine liver samples, the FF and FFPE tissue quality was suitable for all types of analyses (Appendix A). In some cases, the biopsy ruptured into two parts during subsequent FFPE preparation and sectioning (Figure 4a,c and Figure 5a,b), but this still allowed for complete analyses and did not affect the morphological quality of the tissue, as judged by a pathologist. Nevertheless, as the orientation of the biopsy core might slightly change during the transfer from FF to FFPE, precise co-registration between the two sample types is currently challenging. However, due to the small diameter of the biopsy core, it was generally possible to achieve similar cross sections for the FF and FFPE tissues, which worked well in comparative bioanalysis. Limitations regarding the direct comparability of the FF and FFPE tissues were only observed for very small ROIs, and included variances within intra-tumor morphology or the differential expression of IHC markers (Figure 5a,b). To enable comparison between such local intra-tumor heterogeneities, the exact same orientation would be necessary in subsequent FFPE preparations, for which sectioning surfaces would have to be marked during cryo-sectioning. This was not attempted in this study.

For this experiment with clinical tissue, at least 8 sections for both FF and FFPE were prepared, which enabled up to 16 types of analysis for the same biopsy core. In this study, results from MSI, IRI, and H&E staining on the FF-prepared sections, as well as H&E and several IHC stainings of FFPE-prepared sections, were obtained. Reference H&E staining of FF and FFPE tissue suggested that the tumor margins were comparable, despite the better morphological integrity expected of the FFPE tissue (Figure 4d and Figure 5). For the MammaMet sample, nuclear estrogen receptor (ER) was diagnostic in this setting, and its classification as ER-positive mamma metastasis was supported by additional markers including cytokeratin 7 (CK7) (Figure 4d and Appendix A).

Generally, the reference is important to describe the content of the presented section, which could include three possible scenarios: tumor only, as observed for the HCC sample (Figure 5), a mixture of non-tumorous and tumorous tissue, as observed for the MammaMet sample (Figure 4), and non-tumorous regions only, such as those observed in necrosis or cirrhosis. It should be noted that bulk LC-MS/MS omics approaches that are not spatially resolved are often unaware of the percentage of non-tumor tissue in their samples, even though this matter constitutes a significant challenge in tumor marker discovery. This can be avoided by using spatially resolved techniques, such as MSI and IRI, and a pathologist-annotated H&E-stained reference section to confirm tissue types.

In line with the primary aim of this study, namely, to demonstrate the clinical feasibility of the proposed device and workflow, at N = 1, no statement can be made about the potential diagnostic significance of the observed lipid patterns. A comparison of the mass spectra of biopsies prepared using the device with those of resectates prepared without the device and without the use of an embedding medium suggested that no contaminating compounds leached from the inert silicon of the device or the embedding medium (Appendix A). With this in mind, lipid annotation in this study acquired by FT-MRMS in negative ion mode within the *m*/*z* range of 200–2000 with a resolving power of ~135,000 at *m*/*z* 400 used METASPACE (www.metaspace.eu (last accessed on 5 May 2023)) and a false discovery rate (FDR)-controlled algorithm [49]. A search against the LipidMaps (12 December 2017) database resulted in 116 annotations for the mamma metastasis and 4 annotations for the HCC at an FDR of ≤10%, which can be considered level 2 (molecular formula) annotations according to the Metabolomics Standards Initiative [50]. Using discriminant receiver-operator characteristic (ROC) analysis in SCiLS Lab for tumor versus non-tumor ROIs, 54 of the annotated 116 *m*/*z* values in the MammaMet biopsy presented themselves more dominantly in the tumorous region as compared to the non-cancerous tissue. *m*/*z* features that displayed at least a two-fold difference between tumor and non-tumor tissue at <2 ppm mass accuracy are listed in the Appendix A. To showcase the entirety of the workflow, which could be used for the identification of potential biomarkers, an example in-tissue MS/MS fragmentation experiment for *m*/*z* 790.54 was conducted (Appendix A).

As examples of this differential distribution, 2 *m*/*z* features presumably representing 2 different lipids (*m*/*z* 687.545 (±0.008 FWHM), [C38H77N2O6P-H]^−^, 2 candidate molecules with even-numbered fatty acid chains: PE-Cer(d16:1(4E)/20:0) and PE-Cer(d14:1(4E)/22:0)), and *m*/*z* 766.539 (±0.010 FWHM/), [C43H78NO8P-H]^−^, 34 candidate molecules), which were predominantly present in healthy and cancerous regions, respectively, are depicted alongside the pathological reference (Figure 4). Since the HCC biopsies only presented tumorous tissue, no differential analysis of non-tumorous regions was possible. Therefore, only grouping based on their lipid profile in comparison to a database of tumor entities could reveal the relevant molecules for the underlying type of tumor, consistent with other patients with the same diagnosis. Nevertheless, multiple non-annotated *m*/*z* values showed differential intra-tumor distributions (Figure 5). It should be noted that, whereas the proper use of internal standards in untargeted MSI is currently a matter of scientific debate in the MSI community, the validation of any biomarker candidate emerging from this workflow would have to involve its examination by targeted MSI using stable isotope-labeled internal standards [51].

Overall, these results suggest that the proposed device and workflow could be linked with MSI technologies in the clinical analysis of minimal invasive samples including biopsies. There would be no need for an extra sample for analysis performed on FF tissue, thus reducing the risk for the patient. These advances in MSI-based cancer research could potentially pave the way for a more comprehensive clinical study with larger patient numbers, enabling us to achieve more clinically relevant results in the form of a complex lipid pattern database accompanying diagnosis, or a more profound biological understanding of biological processes in future research. However, other spatially resolved techniques such as IRI may also be beneficial in clinical tissue diagnostics [27,29,52]. Using annotated ROIs, e.g., using tumor and non-tumor as labels, spectral differences can be used to guide clinicians in their disease investigations. In IRI, the differential spatial distribution of molecular classes, such as nucleic acids or lipids, is investigated by recording the vibrations of characteristic bonds, e.g., the P-O-vibration in the phosphate backbone of nucleic acids at ~1080 cm^−1^ or the higher abundance of C-O double bonds in lipids at ~1760 cm^−1^ [39]. In IRI, these accumulations of molecular classes can be used to some extent as surrogates of, e.g., higher proliferation or transcription activity in defined ROI, or of disease states that induce the accumulation of lipids, as in steatosis. Therefore, the differential distribution of molecular classes indicated by the characteristic wavenumbers described above was investigated in this study. These coincided well with the pathological annotations for the MammaMet biopsies between tumorous and non-tumorous regions (Figure 4). They also co-localized well with individual *m*/*z* values, suggesting intra-tumor heterogeneity in the HCC sample (Figure 5).

In summary, our results show that combined hyperspectral and histopathological analysis of FF and FFPE tissue on the same biopsy core is feasible for the investigation of biomolecular processes in tumors. The ability to combine these modalities, even on rather small core biopsies, may pave the way for identifying tumor-specific markers or signatures. In addition, answering fundamental questions about clinically relevant biological processes may now be possible, if higher samples numbers for each tumor entity are available. Furthermore, using co-registration between hyperspectral and H&E data in addition to histopathological examinations of the sample should also allow for the implementation of machine learning workflows [53], e.g., for biomarker discovery [54] or effective classification models [55], especially when large-scale studies are performed. Here, the implementation of the presented workflow could be beneficial, since it allows for the routine clinical collection of biopsy samples, which represents a common source of higher sample numbers in the clinic. In addition, it could facilitate early-stage diagnosis using molecular, spatially aware digital pathology, perhaps alongside MS approaches in the operating room, such as intelligent knife technology [56] or the MasSpec Pen [57]. Overall, multimodal correlative imaging using MSI, IRI, and other modalities has been reported by several laboratories [25,48,58]. However, to our knowledge, it has not before been attempted from the same core needle biopsy after consecutive FF and FFPE preparation. The approach presented here suggests that direct correlations between spectral analyses could be achieved using the same biopsy core and therefore be correlated with each other, potentially facilitating a more comprehensive biological understanding via spatial genomics, transcriptomics, proteomics, lipidomics, and metabolomics, which describe the entire downstream process [59].

However, it is important to note that, in clinical assessments where patient outcomes are the first priority, any biopsy tissue must be examined by a trained pathologist prior to research. Thus, for our workflow to be adopted in hospitals, the device would have to be handled by a pathologist.

## 4. Conclusions

In this work, we present a device for the sample collection, freezing, and sample preparation of a core needle biopsy specimen. It enables the reliable and robust preparation of tissue sections for multimodal spatial omics analysis on FF tissues in clinical research and complementary routine histopathology on FFPE tissue from the same longitudinally sectioned biopsy core. Non-rigid co-registration allowed for the data fusion of various spatial omics modalities from the FF tissue for integrative spatially resolved molecular tissue analytics. The proposed approach was successful in an initial clinical feasibility study of five human liver biopsy samples from two different patients with different tumors, a mamma metastasis in the liver and an HCC. For each sample, at least eight adjacent tissue sections with similar shapes and of similar quality were obtained from each preparation type, FF and FFPE. This approach reduces the risk for patients, since no additional biopsies need to be retrieved for analysis requiring FF tissue. Furthermore, the comparability of the multimodal measurement results is greatly improved when using the same biopsy core for all analyses, in contrast to using individual biopsies for each analysis. The generation of FFPE tissue from the same core biopsy also permits direct comparison with histopathology, which is an indispensable diagnostic reference for tissue variants, cancer subtypes, and disease stages. The device and corresponding workflow are cost-efficient, fast, and robust, and may therefore be suitable for standardized implementation in clinical practice. Hence, this standardized workflow appears to be suitable for clinical routine and clinical research into future early-stage diagnosis, prognosis, and targeted therapy.

## 5. Patents

A patent application for the device presented in this study is currently pending.

## Figures and Tables

**Figure 1 cancers-15-02676-f001:**
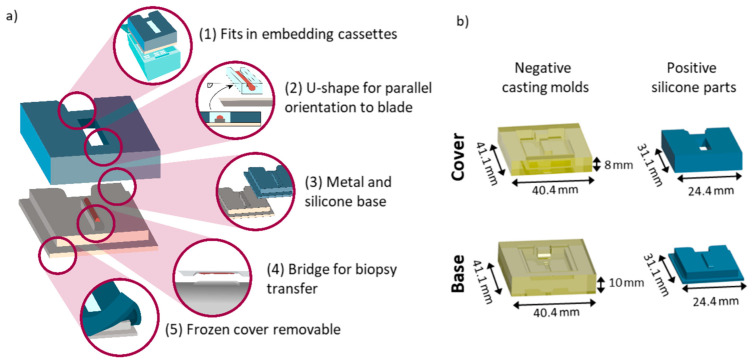
Schematic representation of the features of the biopsy freezing device and conceptual production parts. (**a**) Design choices for the 3D-printed biopsy freezing device, highlighting from top to bottom: (1) the overall sizing fits into commercially available embedding cassettes for the prior labelling (with pencil, aerospace markers, or barcodes) of the sample. (2) A u-shaped mold for the precise orientation of parallel sectioning, plane to blade. (3) Different base materials with different thermal conductive properties tested. (4) A bridge facilitating the transfer of the biopsy from the needle to the device. (5) A silicone cover that is bendable at negative temperatures and therefore removable when frozen. (**b**) Technical drawings of the individually constructed parts of the device for core needle biopsy freezing, including 3D-printed negative casting molds made from resin (yellow) for the cover, as well as the base component and the resulting positive silicone parts (blue) obtained from the negative molds, respectively.

**Figure 2 cancers-15-02676-f002:**
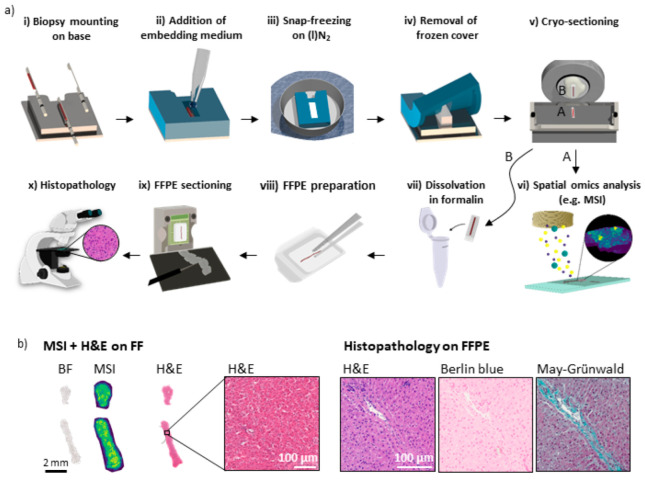
Schematic representation and developmental stage results of workflow for the analysis of fresh-frozen tissue sections and subsequent FFPE tissue sections retrieved from a single needle biopsy core. The workflow combines spectral analysis similar to MALDI MSI and routine clinical histopathology analysis without the need for an additional biopsy. (**a**) First, the preservation of a freshly excised biopsy sample is conducted immediately after needle punching to prevent molecular changes and dislocations in the sample. This process includes (i) the transfer of the biopsy from the needle onto the base of the device, (ii) embedding in a suitable medium after placing the device cover onto the base, (iii) snap-freezing while floating on liquid nitrogen, (iv) when frozen, the removal of the cover (which is bendable at negative temperatures), (v) mounting the sample upside-down on a metal chuck for cryo-sectioning, and mounting sections on suitable slides for the subsequent (vi) multimodal spatial omics analyses, e.g., MS- or FT-IR imaging, requiring FF sample preparation. Following the workflow, the same biopsy is preserved afterwards for clinical routine pathology examination, which includes (vii) the dissolution of the embedding medium and the fixation of the biopsy in formalin, (viii) embedding in paraffin, (ix) FFPE tissue sectioning and mounting on glass slides for subsequent (x) histopathological staining and analyses. (**b**) Exemplary results from bovine livers collected during the device’s developmental stage and the corresponding workflow. From left to right: bright field (BF) image, MS imaging (MSI) and H&E staining all performed on FF sections as well as H&E, Berlin Blue, and May–Grünwald stainings performed on the FFPE sections retrieved from the same biopsy core following the workflow described in (**a**).

**Figure 3 cancers-15-02676-f003:**
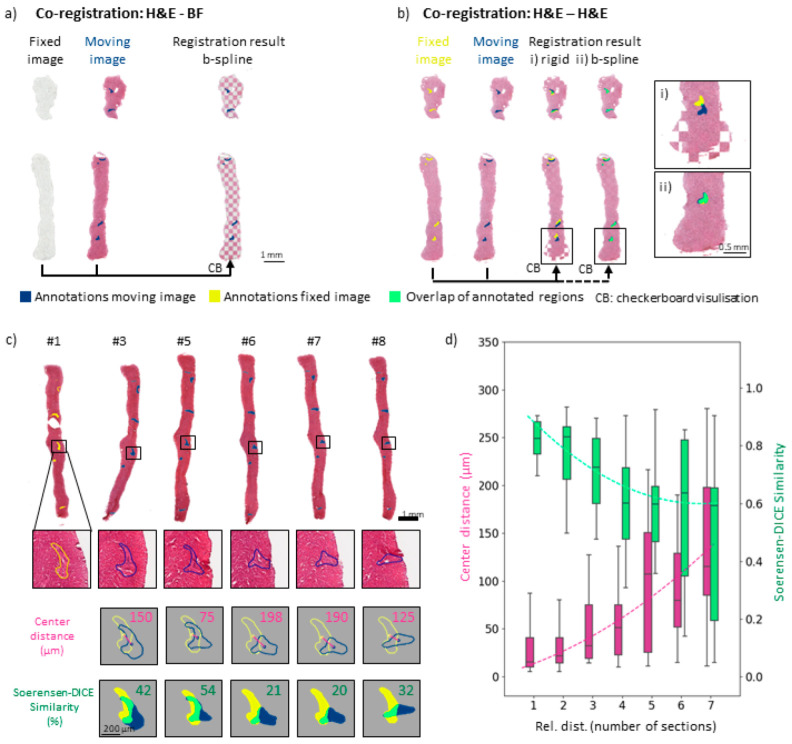
Quality assessment of image co-registration for (FF) bovine liver sections. (**a**) The co-registration of adjacent FF sections, one H&E stained the other a bright field (BF) image, for the subsequent spatial omics reference. The H&E-stained image was deformed as a moving image to fit the fixed BF image, with the final registration result depicted as a checkerboard representation using a non-linear registration method (b-spline). (**b**) The co-registration of two adjacent H&E-stained sections (one fixed, one moving image) with a comparison of linear (rigid) and non-linear (b-spline) registration methods, the later allowing for local deformations. Magnifications show the advantages of local deformation being possible in b-spline registration, thus enabling a more precise fit of the arteries, which were annotated as corresponding ROIs after the registration. (ROIs: fixed section (yellow), moving section (blue), overlap after co-registration (green)) (**c**) Assessment of registration quality over multiple distances between sections, H&E-stained section numbers (#) (sections #2 and #4 were measured by MSI and were not used for the registration quality assessment). From top to bottom (Top) complete cross-sections of biopsy cores. (Middle) magnification of corresponding ROI (artery) for each section. Note that the shape of the artery changes markedly with progression through the biopsy core. (Bottom) parameter scores to assess registration quality, the ROIs’ center of gravity distance score and the Sorensen–Dice coefficient (SDC) overlaying area score. For each distance between sections, the respective section (#3–#8, blue ROIs) was moved to fit the fixed section (#1, yellow ROIs). Scores are depicted in the top right corners individually. (**d**) Increasing center distance scores (pink) and decreasing SDC scores (green) were calculated for all bovine biopsy samples and their six annotated ROIs and plotted over the relative distance between sections (e.g., distance 1 between sections #5 and #6, distance 2 between sections #5 and #7 and so on). Deviations are depicted as box plots (box representing the lower to upper quartiles with median representation, whiskers representing the highest values within the 1.5 × inter-quartile range (IQR). A second-order polynomial function was fitted to the data to guide the eye (depicted as a dashed line).

**Figure 4 cancers-15-02676-f004:**
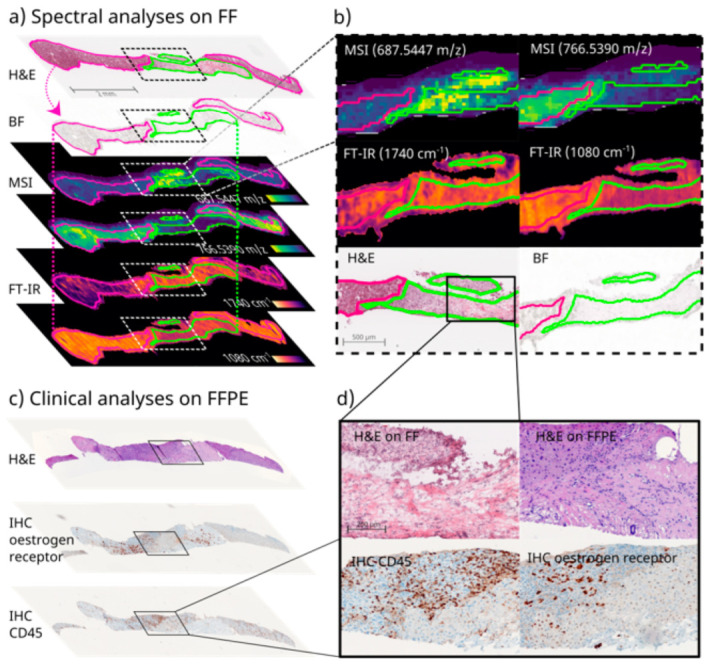
Multimodal imaging of FF and FFPE tissue sections of one biopsy core of a mamma carcinoma metastasis in the liver. Following the proposed workflow, FF as well as FFPE sections could be prepared from the same biopsy core for spectral analyses as well as histopathological stainings. (**a**) Spectral analysis of FF sections including an H&E-stained reference section, a bright field (BF) image for registration, MALDI magnetic resonance MSI (spatial resolution 50 µm, *m*/*z* range 200–2000), and FT-IR imaging (spatial resolution 25 µm, spectral range 750–4000 cm^−1^). Via co-registration (bent dotted arrows), pathologist annotations (normal liver tissue (pink); liver metastasis of mamma carcinoma (green)) for the H&E reference could be transferred (straight dotted lines) onto spectral analysis for the discriminant analysis of *m*/*z* values and wavenumbers. Two examples each of differential distributions between tumorous and non-tumorous regions are depicted, including (**b**) magnifications: For MSI, *m*/*z* 687.545 (±0.008 FWHM) displayed higher ion intensity in the tumor region and *m*/*z* 766.539 (±0.010 FWHM) higher ion intensity in the non-tumorous region. For IRI, 1080 cm^−1^ corresponding to P-O-bonds characteristic of nucleic acids and 1760 cm^−1^ corresponding to C-O-double bonds more abundant in lipids were enriched in tumor and non-tumor regions, respectively. (**c**) Pathological analysis of sections of the same biopsy core after FFPE preparation, including H&E, IHC (estrogen receptor and CD45 (lymphocyte common antigen) as a marker of Kupffer star cells) staining with (**d**) magnifications. Additionally, side-by-side H&E staining is depicted on FF and FFPE sections for a comparison of tissue quality, tumor margins, and the sectioning plane.

**Figure 5 cancers-15-02676-f005:**
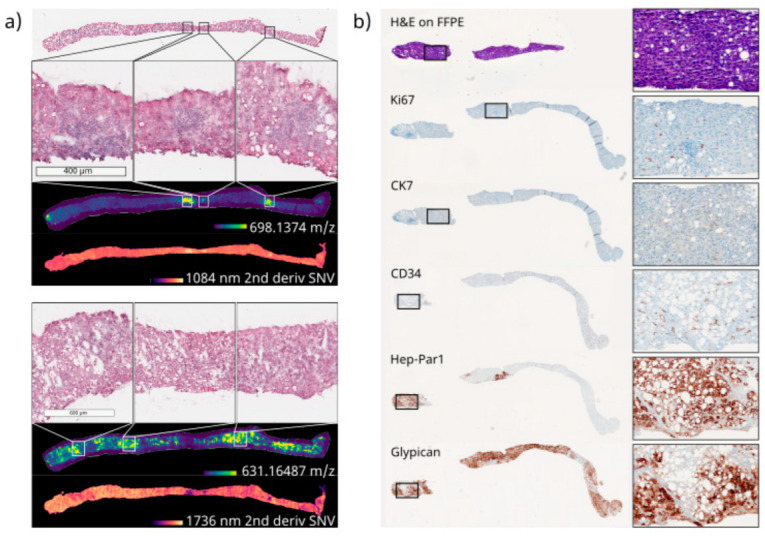
Multimodal imaging of FF and FFPE sections of one biopsy core of an HCC. Following the proposed workflow, FF and FFPE sections could be retrieved from the same biopsy core for spectral analyses, as well as histopathological staining. (**a**) Spectral analysis conducted on FF sections including an H&E-stained reference section, a bright field (BF) image for registration, MALDI MSI (spatial resolution 50 µm, mass range 200–2000 Da, resolution at 400 *m*/*z*~135.000, 800 *m*/*z*~73.500), and FT-IR imaging (spatial resolution 25 µm, spectral range 750–4000 cm^−1^, spectral resolution 4 cm^−1^, pre-processed by second derivative and standard normal variate (SNV)). Pathological annotation suggested the entire biopsy core consisted of HCC tumorous tissue. Nevertheless, different distributions of particular *m*/*z* values and molecular classes (locally increased signal intensities) could be observed, suggesting intra-tumor heterogeneity. Two examples of each differential distribution are depicted as examples, including magnifications of the H&E reference image, to correlate morphological variances with the spectral findings. For MSI *m*/*z* 698.1374 (±0.008 FWHM), *m*/*z* 631.1649 (±0.007 FWHM)), as well as two wavenumbers (1080 cm^−1^, 1736 cm^−1^) indicative of P-O bonds that are more abundant in nucleic acids and C-O double bonds that are more abundant in lipids, are shown beneath the according MSI images. (**b**) Adjacent FFPE sections are retrieved from the same biopsy core after subsequent FFPE preparation and histopathological analysis, which is routinely performed in a clinical context for the precise diagnosis of HCC (CK7, CD34 and glypican3, supported by Ki67 and Hep-Par1 staining); they are depicted to emphasize the suitable tissue quality for routine analysis of FFPE tissue after FF preparation.

## Data Availability

Data are available from the authors upon reasonable request.

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
