# Peer review of "Spatial Omics Imaging of Fresh-Frozen Tissue and Routine FFPE Histopathology of a Single Cancer Needle Core Biopsy: A Freezing Device and Multimodal Workflow"

_cancers, 2023, doi:10.3390/cancers15102676_

Round 1

Reviewer 1 Report

line51: Keywords: keywords are often specific words, not phrases,

line 119: Cazares and co-workers: it is best to write Cazares et al.(year)

line 120: Shiraishi et al.(year)

2.1. Materials: what is described here is not material. It is the working solutions. 

It is 2.2. that must be material

line 150: Human tissue specimen were collected from excised tumors during surgery. In which hospital?

line 150: Human sampling was covered by ethic vote (2012-293N-MA, 20.062012) of the Medical Ethics Committee II of the Medical Faculty Mannheim of Heidelberg University with informed consent of all participants. I propose to put this part and the related parts under the subtitle: ethical considerations.

line 175: solution A : solution B, 1:1 (v:v)): give the composition of the solutions

I have the impression that the figures in the manuscript and the supplementary figures are identical

Author Response

Response: We thank the reviewer for his/her suggestions to improve the manuscript. We are pleased that the reviewer found the references suitable, the study design appropriate and methods, results and conclusion comprehensible.

Comments and Suggestions for Authors:

line51: Keywords: keywords are often specific words, not phrases

Response: Thank you for your suggestion.

Changes: Keywords have been adapted to the following: MALDI imaging, biopsy, multimodal, spatialomics, infrared, immunohisto­chemistry, co-registration, mass spectrometry

line 119: Cazares and co-workers: it is best to write Cazares et al.(year)

Response: Thanks for pointing it out.

Changes: Has been adapted.

line 120: Shiraishi et al.(year)

Response: Thanks for pointing it out.

Changes: Has been adapted.

2.1. Materials: what is described here is not material. It is the working solutions. 

Response: Thanks for pointing it out.

Changes: The materials listed in 2.1. have been incorporated into the respective subsections of the combined chapter 2. Materials and Methods.

It is 2.2. that must be material

Response: Thank you for the suggestion.

Changes: This has been addressed as described in the comment above.

line 150: Human tissue specimen were collected from excised tumors during surgery. In which hospital?

Response: The study was conducted within the Forschungscampus M²OLIE wherefore all clinical samples were excised during surgery performed at the University Clinic in Mannheim.

Changes: Information has been added to the manuscript.

line 150: Human sampling was covered by ethic vote (2012-293N-MA, 20.062012) of the Medical Ethics Committee II of the Medical Faculty Mannheim of Heidelberg University with informed consent of all participants. I propose to put this part and the related parts under the subtitle: ethical considerations.

Response: Thank you for the suggestion.

Changes: Details of ethics approval have been moved to the “Informed Consent Statement“ at the end of the manuscript.

line 175: solution A : solution B, 1:1 (v:v)): give the composition of the solutions

Response: The above mentioned solutions refer to the two-component system of the silicone supplier Silconic, which are subject to the company's confidentiality. Therefore, in this study, only the commercially available components, named by the manufacturer as solution A and solution B, can be purchased and used as indicated for manufacturing the proposed device.

Changes: The information of the manufacturer providing a two-component system for silicone mixture is now also provided in the manuscript.

I have the impression that the figures in the manuscript and the supplementary figures are identical

Response: Figures which did not present key findings but were relevant for the study design or individual data analysis were only featured in the supplementary information. For example, the accuracy estimation of serial sections from an individual biopsy is only shown in the supplementary information for each of the replicates. Only here, outliers of single deformed sections can be observed. This information is lost when calculating an average accuracy over multiple serial sections of several biopsy replicates. Thus, in conclusion only a comprehensive understanding of whether or not the registration is qualified for this particular use case is required, a summary figure including all the biopsies tested and therefore averaged measures of the registration accuracy, are shown as one of the figures in the main manuscript. For this reason, some similarity of figures in the main manuscript and the supplement is observable.

Changes: A clearer statement was made differentiating these figures and their individual value in the main manuscript. “These individually calculated registration accuracy statistics allow a detailed assessment of each individual section and outliers of the manual sectioning process. In contrast, a summary statistic as depicted in Figure 3d, which enables a superordinate evaluation of the precision of co-registration across several replicates consistently suitable for the study under consideration.”

Reviewer 2 Report

The authors present data regarding the construction of a device for use in a diagnostic setting combining omic imaging approaches and histology. The authors have created a device for MSI / IRI on fresh-frozen samples that can be combined with histology using FFPE samples.

This reviewer found the study both interesting and important. However, in the author’s own words they state:

1. Line 135-136: “This study presents the design of the device and the 3D-printing manufacturing process. The corresponding workflow is described in detail.”

2. Line 651-652: “we present a device for sample collection, freezing and sample preparation of core needle biopsy specimen.”

Considering the points above, and the fact that any biological data is more of a proof of principle that the engineering works rather than driving new biological insights, it is this reviewers’ opinion the manuscript is more related to an engineering question rather than cancer biology. This is obviously an editorial decision but this reviewer does not feel qualified to comment on the engineering focus and suggests that the journal is not a good fit for this study.

Author Response

Response: We are pleased that you found the references to be suitable, and the methods, results and conclusion comprehensible. Subsequent comments and suggestions have been addressed individually as described below.

Comments and Suggestions for Authors

The authors present data regarding the construction of a device for use in a diagnostic setting combining omic imaging approaches and histology. The authors have created a device for MSI / IRI on fresh-frozen samples that can be combined with histology using FFPE samples.

This reviewer found the study both interesting and important. However, in the author’s own words they state:

  1. Line 135-136: “This study presents the design of the device and the 3D-printing manufacturing process. The corresponding workflow is described in detail.”
  2. Line 651-652: “we present a device for sample collection, freezing and sample preparation of core needle biopsy specimen.”

Considering the points above, and the fact that any biological data is more of a proof of principle that the engineering works rather than driving new biological insights, it is this reviewers’ opinion the manuscript is more related to an engineering question rather than cancer biology. This is obviously an editorial decision but this reviewer does not feel qualified to comment on the engineering focus and suggests that the journal is not a good fit for this study.

Response: We thank reviewer 2 for considering our “study both interesting and important”.

It is true that it was not our intention to provide new biological insights here. However, we submitted this study to “Cancers” for the Special Issue on MSI on purpose, since clinical cancer researchers typically don’t read engineering journals. We feel that it is very important that the device and workflow presented here be known by, used by (and the paper cited by) clinical researchers who examine cancer biology. We agree that it is, of course, an editorial decision.

That said, beyond the technical innovation of the proposed device and corresponding workflow, clinical suitability has also been shown in this proof of concept study. Thus, we would like to see this study as valuable and important “Advances in Mass Spectrometry Imaging-Based Cancer Research”. An effort was made to make the manuscript comprehensible to a wider audience and enable an evaluation also from a clinical point of view. This is also reflected by the opinion of reviewer 3, who stated: “It is an article that, without presenting sensational results, demonstrates and achieves the proposed objectives, with the analysis of tumor biopsies in the necessary proof of concept.”

Changes: Follow up experiments were conducted to further show the suitability of the proposed workflow for clinical diagnosis in cancer research. Briefly, identification of individual m/z values were pursued by MS/MS fragmentation and added to chapter 3.4. Clinical proof-of-concept of FF and FFPE-tissue multimodal spatial omics analysis on the same needle biopsy core. This covers the entire MSI workflow from best available sample preparation, to spectral analysis, to follow-up experiments for structural elucidation of biomolecules corresponding to m/z values. This completes in our opinion – in a proof-of-concept kind of way – an outline for future clinical research to the point of potential biomarker discovery.

Reviewer 3 Report

In the manuscript “Spatial omics imaging of fresh-frozen tissue and routine FFPE 2 histopathology on a single cancer needle core biopsy: freezing 3 device and multimodal workflow” the authors combine the use of several techniques in the analysis of the same needle biopsy tissue sample, as the traditional ones histopathological examination in formalin-fixation and paraffin-embedding (FFPE) samples with molecular analysis by mass spectrometry imaging (MSI) and infrared spectroscopy imaging (IRI) in fresh-frozen samples.

A very technical, well-organized and well-written manuscript, with enough details to allow even the non-specialist to understand and envision the advantages of integrating information and its potential advantages in clinical evaluation. It is an article that, without presenting sensational results, demonstrates and achieves the proposed objectives, with the analysis of tumor biopsies in the necessary proof of concept.

Author Response

Response: Thank you for your evaluation of our manuscript. We are pleased that you found the references suitable, the study design appropriate and that you found the methods, and results comprehensible.

Comments and Suggestions for Authors

In the manuscript “Spatial omics imaging of fresh-frozen tissue and routine FFPE 2 histopathology on a single cancer needle core biopsy: freezing 3 device and multimodal workflow” the authors combine the use of several techniques in the analysis of the same needle biopsy tissue sample, as the traditional ones histopathological examination in formalin-fixation and paraffin-embedding (FFPE) samples with molecular analysis by mass spectrometry imaging (MSI) and infrared spectroscopy imaging (IRI) in fresh-frozen samples.

A very technical, well-organized and well-written manuscript, with enough details to allow even the non-specialist to understand and envision the advantages of integrating information and its potential advantages in clinical evaluation. It is an article that, without presenting sensational results, demonstrates and achieves the proposed objectives, with the analysis of tumor biopsies in the necessary proof of concept.

Response: Thank you very much for this very positive evaluation of our manuscript. We are pleased that you found it well-organized and well-written so that the emphasis on the advantages for clinical evaluation in the field of cancer research could be conveyed.

Changes: Smaller changes in accordance with comments from other reviewers have been made, to further improve comprehensibility of the proposed study. For details, please refer to the manuscript.

Reviewer 4 Report

Title:  Spatial omics imaging of fresh-frozen tissue and routine FFPE2 histopathology on a single cancer needle core biopsy: freezing3 device and multimodal workflow

Author:  Rittel, Miriam F. et.al.,

Description of the work:  This paper details production and use of a new devise that enables analysis of needle biopsies by high content modalities such as imaging mass spectrometry (IMS) on non-fixed slices and standard microscopy on FFPE slices.

Major Comments:

1.  The need for accurate and fast analysis of needle biopsy samples by high content modalities and by standard microscopic imaging technologies is clear.  The data presented support the claim that the devise shown and the protocols for using that devise can effectively produce both types of data from adjacent slices of a single needle biopsy.   

Minor Comments:

1. The authors make too much use of the convention of putting text into parentheses that they wish to be considered as a separate idea or an aside.  This convention is over used and the ideas that are being presented as asides by placing them in parentheses could be better conveyed by more careful writing in standard format.

2.  The term “golden standard” is used on ln 24 and ln 102.  The usual term is “gold standard”.

3.  ln 71. “fashion7” has a double period after it.

4.  ln 112 & 113.  Consider “procedure that limits” instead of “procedure limiting” and consider “error” instead of “error sources”.

5.  ln 200.  “as freely”  should be “a freely”.

6.  ln 249.  “ FF biopsies”  should be “FF biopsy”.

7.  ln 310.  “Learnings made”  would be better expressed as “Lessons learned”

8.  ln 330.  “onto the bride” Please check that this is what is intended.

9.  ln 366 and other places.  “subsequent tissue sections”  is not sufficiently clearly defined.  Subsequent can be interpreted as being taken at a later time.  If that is what is intended then please be clear.  If, however, the intent is “adjacent tissue sections” then please use terminology that is not subject to misinterpretation.

Author Response

Response: Thank you for your evaluation of our manuscript. We are pleased that you found the references suitable, the study design appropriate and that you found the methods, results and conclusion comprehensible.

Changes: Linguistic improvements have been incorporated into the manuscript.

Comments and Suggestions for Authors

Title:  Spatial omics imaging of fresh-frozen tissue and routine FFPE2 histopathology on a single cancer needle core biopsy: freezing3 device and multimodal workflow

Author:  Rittel, Miriam F. et.al.,

Description of the work:  This paper details production and use of a new devise that enables analysis of needle biopsies by high content modalities such as imaging mass spectrometry (IMS) on non-fixed slices and standard microscopy on FFPE slices.

Major Comments:

  1. The need for accurate and fast analysis of needle biopsy samples by high content modalities and by standard microscopic imaging technologies is clear. The data presented support the claim that the devise shown and the protocols for using that devise can effectively produce both types of data from adjacent slices of a single needle biopsy.   

Response: Thank you for your evaluation. Thank you for your assessment that there is a need for a device and workflow that we present and that the device can help produce both types of data from the same needle biopsy.

Minor Comments:

  1. The authors make too much use of the convention of putting text into parentheses that they wish to be considered as a separate idea or an aside. This convention is over used and the ideas that are being presented as asides by placing them in parentheses could be better conveyed by more careful writing in standard format.

Response: Thank you for pointing this out.

Changes: All such parentheses have been removed and their content is now included in the main text.

  1. The term “golden standard” is used on ln 24 and ln 102. The usual term is “gold standard”.

Response: Thank you for pointing this out.

Changes: In both cases, this has been adapted in the manuscript.

  1. ln 71. “fashion7” has a double period after it.

Response: Thank you for noting.

Changes: Has been adapted in the manuscript.

  1. ln 112 & 113. Consider “procedure that limits” instead of “procedure limiting” and consider “error” instead of “error sources”.

Response: Thank you for the suggestions.

Changes: Both have been adapted in the manuscript.

  1. ln 200. “as freely”  should be “a freely”.

Response: Thank you for noting.

Changes: Has been adapted in the manuscript.

  1. ln 249. “ FF biopsies”  should be “FF biopsy”.

Response: Thank you for the suggestion.

Changes: It has been adapted in the manuscript to “After cryosectioning, the remainder of each FF biopsy in embedding medium was immersed in 5 mL of 10% phosphate-buffered formalin…” 

  1. ln 310. “Learnings made”  would be better expressed as “Lessons learned”

Response: Thank you for noting.

Changes: This was adapted in the manuscript.

  1. ln 330. “onto the bride” Please check that this is what is intended.

Response: Thank you for pointing out this typo, which we corrected. Moreover, the term “bridge” was used in this publication for the elevation present on the base plate since it looks like the bridge on a stringed instrument, like a guitar.

Changes: To make this point more comprehensible, an explanation of the “bridge” being the elevation on the base compartment was added to the above mentioned paragraph.

  1. ln 366 and other places. “subsequent tissue sections”  is not sufficiently clearly defined.  Subsequent can be interpreted as being taken at a later time.  If that is what is intended then please be clear.  If, however, the intent is “adjacent tissue sections” then please use terminology that is not subject to misinterpretation.

Response: Thank you for pointing out this possible misunderstanding. Indeed, in this publication “subsequent” and “adjacent” were used synonymously.

Changes: For better understanding, the word “adjacent” is now used throughout the entire manuscript. 

Reviewer 5 Report

General review: Rittel et al. are describing an approach to help bridge the gap between histopathology workflows and to MSI workflows by introducing a device for needle biopsies that allows both samples preparations for MSI and histopathology. This device allows more information to be obtained from a single biopsy. A proof-of-concept experiment included a liver cancer specimen that was flash-frozen for MSI and IRI followed by FFPE processing for clinical pathology analysis.

The major point to address is that in general assessment the tissue will first be given to pathology prior to research, thus this workflow may not be adopted in many hospitals. As the patient outcome is number one, pathology will always have the first pass of the tissue. The device is a nice result, however a mention of this is needed. It would be too dangerous for a pathologist to conclude with researchers handling tissue first, who do not have the proper training. Unless you see this device with the pathologist, I think a discussion point needs to be mentioned.  

Comments:

1.      On lines 103-105 the authors state that frozen sections are needed for MSI, however several publications have shown success in using MSI from FFPE tissue for small molecule analysis. For example PMID: 27414759 should be cited.

2.      There are no control tissues that were tested before the clinical samples, there is an assumption here that the workflow is correct, very little was done with bovine control tissue.

a.      I would like to see controls, including, is there any leaching of solvents from the 3D mold?

b.      I would like to see IHC/or MSI from both the FFPE and flash frozen tissue to show that there were no additional artifacts in either workflow. It is too premature to go straight into clinical samples without proper controls. Or taking a control tissue for FFPE and the FF to FFPE treatment to see if there are any changes.

c.       Does the embedding medium have any effect on the signal? A spectrum of the embedding medium would be helpful to show that it does not affect the signal.

d.      How is the FFPE tissue being affected by dissolving the FF embedding medium? There are no controls here. I would like to see control tissue with and without this step to show that there is no effect.

3.      Not sure why the FF to FFPE is needed; it is well accepted to perform IHC on FF tissues. Could the authors add to this

MSI comments:

4.      For the MSI work, no mention of tissue normalization, as there are clinical specimens that is the confidence that the signal visualized is not an artifact of ion suppression and different ionization efficiencies based on tissue heterogeneity. It is well known that ion signal is varied due to cell density and would show biased images if not normalized. Was there no internal standard sprayed onto the tissue? Also, no mention of how the MALDI matrix was sprayed. For example, in Figure 4, poor normalization could show that the 687 m/z is higher in the non-tumor region; however, with proper normalization, this could be the inverse. 

a.      I would like to see a labeled lipid standard sprayed onto the tissue and the ion images normalized to that signal.

5.      How are you accounting for ion suppression?

6.      There are limited annotations in this paper. You cannot say that a m/z is linked to non-tumor or tumor regions if they are not annotated. These signals could come from solvents, matrix-related peaks. No biological conclusions can be made with an m/z value. There is a need for MSMS or an orthogonal method to provide annotations since these are clinical samples and statements were made.

7.      Additionally, the authors mention small molecules throughout the paper but do not show any ion images. The DAN MALDI matrix shows very nice TCA, lactate, pyruvate, purines, and other compounds and would significantly improve the manuscript. 

8.      As there was a large effort done on image registration, it would be nice to see some overlays of MSI and IHC (not H&E and MSI, this has been done several times and not novel)  and have some conclusions based on molecular signatures.  

Author Response

see attached document.
